# Prevalence and genetic variability of occult hepatitis B virus in a human immunodeficiency virus positive patient cohort in Gondar, Ethiopia

**Nishi H. Patel**[1], **Vanessa Meier-Stephenson**[1], **Meaza Genetu**[2], **Debasu Damtie**[2,3,4], **Ebba Abate**[2,5], **Shitaye Alemu**[6], **Yetework Aleka**[2], **Guido Van Marle**[1], **Kevin Fonseca**[1,7], **Carla S. Coffin**[1,8], **Tekalign Deressa**[2,5]*

1 Department of Microbiology, Immunology and Infectious Diseases, Cumming School of Medicine, University of Calgary, Calgary, Alberta, Canada, 2 Department of Immunology and Molecular Biology, School of Biomedical and Laboratory Sciences, College of Medicine and Health Sciences, University of Gondar, Gondar, Ethiopia, 3 Food Animal Health Research Program, CFAES, Ohio Agricultural Research and Development Center, Department of Veterinary Preventive Medicine, The Ohio State University, Wooster, OH, United States of America, 4 Global One Health LLC, Eastern African Regional Office, Addis Ababa, Ethiopia, 5 Ethiopian Public Health Institute, Addis Ababa, Ethiopia, 6 School of Medicine, College of Medicine and Health Sciences, University of Gondar, Gondar, Ethiopia, 7 Provincial Laboratory for Public Health, Alberta Health Services, Calgary, Alberta, Canada, 8 Division of Gastroenterology and Hepatology, Department of Medicine, Cumming School of Medicine, University of Calgary, Calgary, Alberta, Canada

* tekalign09@gmail.com

**Data Availability Statement:** All relevant data are within the manuscript

## Abstract

### Background

Occult hepatitis B (OHB) is a major concern in HIV infected patients as it associates with a high risk of HBV reactivation and disease progression. However, data on the prevalence of OHB among HIV positive patients in Ethiopia is lacking. This study aims to determine the prevalence of OHB in HBV/HIV co-infected patients from Gondar, Ethiopia.

### Methods

A total of 308 consented HIV positive patients were recruited from the University of Gondar Teaching Hospital, Ethiopia. Clinical and demographic data of the participants were recorded. Plasma was tested for HBsAg and anti-HBc using commercial assays (Abbott Architect). In HBsAg negative anti-HBc positive patient samples, total DNA was isolated and amplified using nested PCR with primers specific to HBV polymerase, surface and pre-core/core regions, followed by Sanger sequencing and HBV mutational analysis using MEGA 7.0.

### Results

Of the total study subjects, 62.7% were female, median age 38.4 years, interquartile range (IQR): 18–68, and 208 (67.5%) had lifestyle risk factors for HBV acquisition. Two hundred and ninety-one study subjects were HIV+/HBsAg-, out of which 115 (39.5%) were positive

**Funding:** The authors received no specific fund for this study.

**Competing interests:** The authors have declared that no competing interests exist.

for anti-HBc. Occult hepatitis B was detected in 19.1% (22/115) of anti-HBc positive HIV patients. HBV genotype D was the predominant genotype (81%) among OHB positive patients. Mutations associated with HBV drug resistance, HBV reactivation, and HCC risk were detected in 23% (5/22), 14% (3/22) and 45.5% (10/22) of patients, respectively.

## Conclusion

This study found a high rate of occult hepatitis B in HIV patients. Further, high rates of mutations associated with HBV reactivation, drug resistance, and HCC risk were detected in these patients. These data highlighted the need for integrating OHB screening for proper management of liver diseases in HIV patients.

## Introduction

Occult hepatitis B (OHB) has been increasingly recognized over the last 2 decades as a public health concern. It is characterized by the presence of hepatitis B virus (HBV) DNA in plasma, liver, and/or peripheral blood mononuclear cells (PBMC) of patients with no detectable hepatitis B surface antigen (HBsAg) in serum [1]. Occult HBV infection has been associated with the development of hepatocellular carcinoma (HCC) [2–4]. OHB was detected in tumour tissue of HBsAg negative HCC patients with prevalence of 30% to 60% [2–5]. Further, it was detected in serum and/or liver of patients with chronic hepatitis of unknown origin with the prevalence ranging from 19%-31% [6, 7]. In these patients, OHB infection associated with severe liver damage and progression of the liver lesion to cirrhosis.

Occult hepatitis B infection is common in HIV infected patients [1, 8, 9]. A study by Coffin et al., for instance, has shown 17.8% and 40% prevalence rates of OHB in serum and PBMC of HIV infected patients respectively [8]. Similar study in a cohort of HIV infected people reported 47% prevalence rate of OBH [9]. In HIV patients, OHB is associated with adverse clinical outcomes including high rate of hepatotoxicity induced by ART, higher risk of hepatic diseases, faster progression of HIV infection, and reactivation of OHB infection. Moreover, as several antiretroviral drugs (ARVs) have dual anti-HIV and anti-HBV activity, there is high possibilities of selecting for resistance mutations in HBV [10–12]. Thus, determining the epidemiology of OHB among HIV patients could significantly impact clinical management of these infections.

The burden of HBV is particularly high in low and middle-income countries, such as Sub-Saharan Africa [13, 14]. However, most people infected with this virus remain unaware of their status and are at an increased risk of liver-related morbidity and mortality. In Ethiopia, there is no report on the epidemiology of occult HBV infections in people living with HIV (PLHIV). This study aims to estimate the prevalence of OHB in a cohort of HIV infected patients in Northwest Ethiopia. Further, we also determined the genotype of HBV, and mutations associated with HCC risk, HBV reactivation, and drug resistance.

## Materials and methods

### Study population and setting

This cross-sectional study was conducted at the University of Gondar Teaching Hospital. The hospital provides in-patient and outpatient medical service to ~5 million people in Northwest Ethiopia. In total, 308 consented HIV-1 positive patients were recruited from March 2016 to

July 2016 from an outpatient antiretroviral (ART) clinic at the University of Gondar Teaching Hospital. We used single population proportion to determine the sample size as described previously [15]. Participants with end-stage acquired immunodeficiency syndrome (AIDS), multiple illness, immunosuppression, and severe malnutrition were excluded.

The sociodemographic data (i.e., age, sex, risk factors for HIV or HBV, education) and clinical data (i.e., antiretroviral treatment, CD4+ T cell count, platelet count, and risk of liver disease) were collected through chart review and structured questionnaire. Whole blood was drawn from all the participants. Plasma and PBMCs were isolated using Ficoll-Hypaque gradient method. The plasma and PBMCs samples were transported to the University of Calgary with appropriate transportation permits from the Public Health Agency of Canada and the University of Calgary Occupational Health and Safety office.

## Sample processing and detection of OHB

The plasma samples were tested for HBsAg and antibody to hepatitis B core antigen (anti-HBc) at the Alberta Provincial Laboratory using commercial assays (Abbott Architect). In HBsAg negative and anti-HBc positive samples, total DNA was isolated from 500μL plasma using standard phenol-chloroform extraction method. A parallel mock (phosphate-buffered saline) as the negative control was included in the extraction experiments. HBV DNA was amplified using in-house nested PCR using primers specific for HBV surface (S), polymerase (P), and/or pre-core/core (pre-C/C) regions. The HBV S and P regions were amplified using `TGCTGCTATGCCTCATCTTC` and `CARAGACARAAGAAAATTGG` (409 bps), and `CAAGGT ATGTTGCCCGTTTGTCC` and `GGYAWAAAGGGACTCAMGATG` (341 bps). The HBV pre-C/C regions were amplified using `GCATGGAGACCACCGTGAACG` and `GAGGGAGTTCTTCTTC TAGG` (780 bps) and `TCACCTCTGCCTAATCATC` and `GGAGTGCGAATCCACACTCC` (462 bps). The PCR products were confirmed on agarose gel, extracted using Qiagen Gel Extraction Kit (Qiagen, Hilden, Germany), and used for sequencing.

## Sequencing and phylogenetic analysis

HBV mutants and genotype were determined by bidirectional Sanger sequencing of the HBV S, P, and pre-C/C gene fragments. HBV genotype was determined using the NCBI genotyping tool (https://www.ncbi.nlm.nih.gov/projects/genotyping). Phylogenetic and mutational analysis were performed using MEGA 7.0 with Clustal W alignment. Maximum likelihood trees were constructed with the Kimura 2 parameter model with gamma distribution using 1,000 replicates for the bootstrap analysis [16].

## Data analysis

Data analysis was performed using SPSS software (v. 20, SPSS Inc., Chicago, IL). Descriptive statistics such as frequency, mean, and median with interquartile range (IQR) were used to summarize baseline characteristics of the study participants. P-values less than 0.05 were considered statistically significant for all analysis.

## Ethics statement

This study was performed according to the Declaration of Helsinki and received ethics approval from the institutional review board of University of Gondar, and Federal Ministry of Science and Technology of Ethiopia (IRB no. 05/254/2017). All subjects provided written informed consent to participate.

## Results

### Sociodemographic and clinical data

Three hundred and eight (308) consented HIV sero-positive participants were enrolled to this study. Age of the participants was between 18 years to 68 years, with median age 38 years, inter-quartile range (IQR) 27–49 years. About 63% (193) of the participants were female, and 67% (208) had lifestyle risk factors (i.e., such as tattooing, unsafe injections, and multiple sex partners). Most study subjects (94%) were on combination ART therapy, i.e., Zidovudine (AZT)-lamivudine (3TC)-nevirapine (NVP) or Tenofovir Disoproxil Fumarate (TDF)-3TC-Efavirenz (EFV) (Table 1).

### Prevalence of OHB and HBV genotype

Out of 308 HIV infected study participants, we previously reported that 17 (5.5%) patients were chronically infected with HBV (HBsAg+) [15]. In this study, we evaluated the prevalence of OHB among 291 HBsAg sero-negative patients in the same cohort. One hundred and fifteen (115) out of 291 (39.5%) HBsAg negative persons were anti-HBc positive. Occult hepatitis B (HBV DNA) was detected in 22/115 (19%) anti-HBc positive patients (Fig 1).

The baseline characteristics of OHB positive HIV patients were presented in Table 2. The median age of OHB positive patients was 40 years, IQR 24–56 years. Most of the patient (20/22) were on combination ART with 3TC and/or TDF. The median CD4+ T cell and platelet count in OHB patients was 330 cells/µL (IQR 343.5, range 6.6–1051) and 291 cells/µL (IQR 88.3, range 196–474), respectively.

### HBV genotyping and mutational analysis

In this study, we were able to PCR amplify and sequence all 22 occult hepatitis B positive HIV patients. Hepatitis B virus genotype D was predominant among OHB positive persons (18/22, 81%). HBV genotypes E, A, and C were also detected in 2/22 (9%), 1/22 (5%), and 1/22 (5%), respectively (Table 3). These HBV genotype results are supported by the phylogenetic analysis (Fig 2).

Through analysis of HBV S region sequences, mutations associated with HBV reactivation (L175S, G185E, S204R) were detected in 3/22 (14%) patients. Further, anti-HBV drug resistant mutations (rtI169T, rtV173L, rtA181T, rtT184A/C/G, rtA194T, rtS202I, M204I, rtM250L/V) were detected in 5/22 (23%) patients. These mutations were associated with 3TC, entecavir, and/

**Table 1. Summary of the sociodemographic and clinical data of the 308 HIV enrolled patients.**

| Characteristics | Values |
|---|---|
| Median Age (IQR, range), years | 38 (11.0, 18–68) |
| Sex | |
| Male (n, %) | 115 (37.3) |
| Female (n, %) | 193 (62.7) |
| Lifestyle Risk Factors (n, %) | 208 (67.5) |
| Education | |
| No formal education (n, %) | 132 (42.9) |
| History for Liver Disease (n, %) | 7 (2.3) |
| ART (n, %) | 290 (94.2) |
| CD4+ T Cell Count (Median, IQR), cells/µL | 405 (75–734) |
| Platelet Count (Median, IQR), cells/µL | 269 (165–373) |

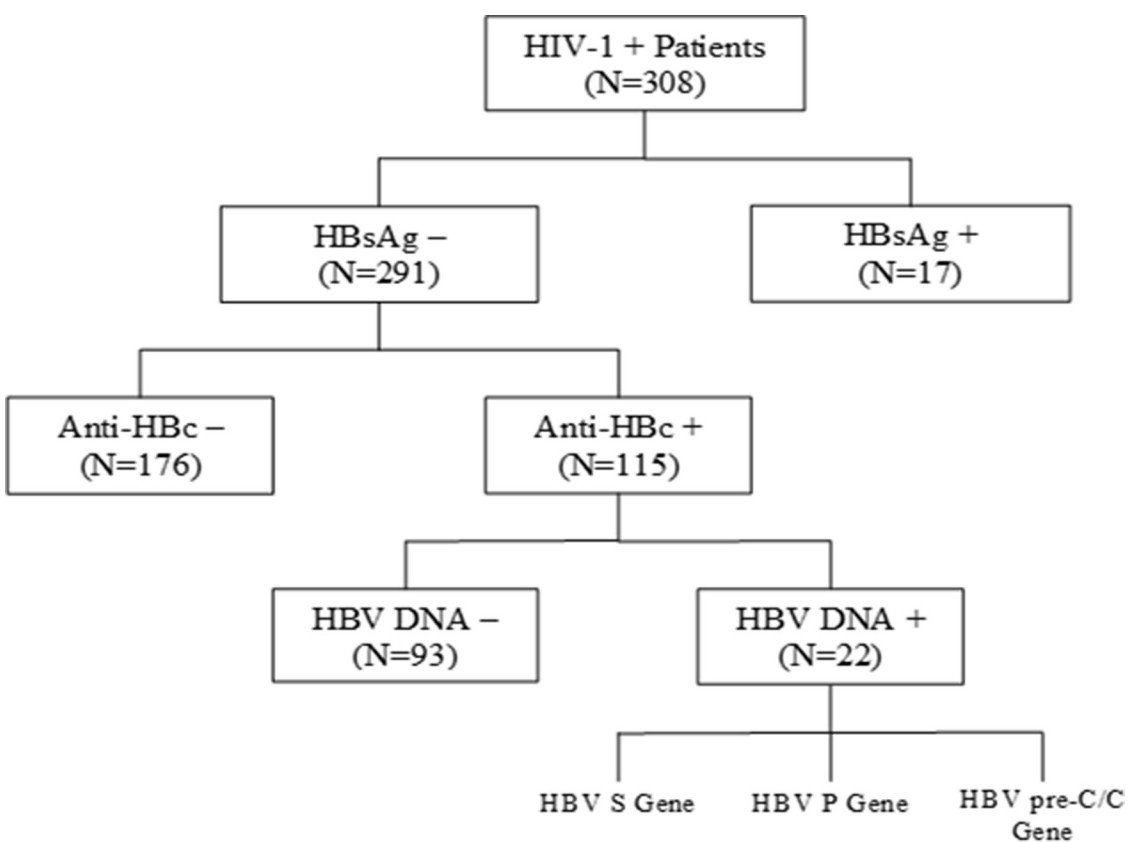

**Fig 1. A flow chart for identification of occult hepatitis B in a cohort of HIV-1 positive patients from March-July 2016 in Gondar, Ethiopia.**

or TDF resistance [16–19]. Of note, mutations for 3TC and entecavir resistance were detected in a patient who had not received ART and anti-HBV therapy. As well, mutation for TDF resistance (rtA194T) was detected in one patient who was on 3TC therapy for 6 years. Mutations associated with liver cirrhosis and HCC risk (HBV pre-core/core region) were detected in 10/22 (45%) patients. The HBV pre-core/core mutation, W28*, was detected in 1/10 patients. Whereas, HBV core mutations (A80I/T/V, E77Q, F24Y, E64D, L116I) were detected in 9/10 patients (Table 3).

## Discussion

Hepatitis B virus and HIV co-infection are common in endemic regions such as sub-Saharan Africa. Occult hepatitis B is a public health concern in HIV patients as it is clinically implicated

**Table 2. Summary of the sociodemographic and clinical data of the 22 OHB/HIV patients.**

| Variables | Values |
|---|---|
| Median Age (IQR, range), years | 40 (16, 27–58) |
| Sex | |
| Male (n, %) | 11 (50.0) |
| Female (n, %) | 11 (50.0) |
| ART (n, %) | 20 (90.9) |
| Median CD4+ T Cell Count (IQR, range), cells/μL | 330 (343.5, 6.6–1051) |
| Platelet Count (IQR, range), cells/μL | 291 (88.3, 196–474) |

**Table 3. Clinical and virological characteristics of OHB/HIV patients (N = 22).**

| Patient ID | Age/ Sex | CD4+ T cell count (cells/μL) | PLT count (cells/μL) | Years on ART | Anti-HBV agent | Mutations | | | HBV Genotype |
|---|---|---|---|---|---|---|---|---|---|
| | | | | | | HBV Reactivation Risk | Drug Resistance | HCC Risk | |
| HP17* | 39/M | 14 | 474 | NA | ----- | L175S | rtM250L, rtM204I | ----- | D |
| HP19 | 42/F | 487 | 305 | 6 | 3TC | S204R, L175S, G185E | rtM250L, rtA181T, rtI169T, rtA194T | UQ | A |
| HP26* | 30/F | 500 | 234 | NA | ----- | ----- | ----- | ----- | D |
| HP61 | 36/F | 88 | 293 | 7 | 3TC | S204R, L175S, G185E | rtM250L, rtM250V | UQ | D |
| HP66 | 50/M | 215 | 289 | 7 | TDF, 3TC | UQ | UQ | A80/I/T/V | D |
| HP73 | 45/F | 53 | 196 | 3 | 3TC | UQ | UQ | E77Q, A80/I/T/V | D |
| HP83 | 41/F | 6.6 | 325 | 9 | 3TC | UQ | UQ | WT | D |
| HP108 | 38/M | 284 | 231 | 8 | 3TC | UQ | UQ | UQ | D |
| HP128 | 33/M | 28 | 196 | 8 | TDF, 3TC | WT | WT | UQ | E |
| HP210 | 27/F | 263 | 261 | 5 | 3TC | WT | rtV173L, rtS202I, rtT184G, rtT184A | UQ | D |
| HP215 | 58/F | 943 | 273 | 7 | 3TC | WT | rtT184G, rtT184C | F24Y, A80/I/T/V | D |
| HP221 | 32/M | 333 | 395 | 2 | TDF, 3TC | UQ | UQ | F24Y, A80/I/T/V | D |
| HP230 | 31/F | 1051 | 293 | 6 | TDF, 3TC | UQ | UQ | W28* | D |
| HP260 | 50/F | 304 | 353 | 10 | 3TC | UQ | UQ | E64D | E |
| HP271 | 32/M | 607 | 272 | 10 | 3TC | UQ | UQ | WT | D |
| HP280 | 27/F | 712 | 285 | 3 | TDF, 3TC | UQ | UQ | A80/I/T/V | D |
| HP287 | 54/M | 424 | 281 | 6 | 3TC | UQ | UQ | WT | D |
| HP290 | 41/M | 776 | 364 | 7 | TDF, 3TC | UQ | UQ | WT | D |
| HP294 | 45/M | 227 | 356 | 1 | TDF, 3TC | UQ | UQ | WT | D |
| HP313 | 25/F | 465 | 413 | ----- | 3TC | WT | WT | E64D, A80/I/T/V | C |
| HP320 | 47/M | 373 | 295 | ----- | 3TC | WT | WT | L116I | D |
| HP323 | 58/M | 327 | 267 | ----- | TDF, 3TC | WT | WT | F24Y | D |

PLT, platelet count (x 10³/mL); ART, anti-retroviral therapy; 3TC, Lamivudine; TDF, Tenofovir Disoproxil Fumarate; NA, not applicable;

*, ART naive; WT, wild type; UQ, HBV DNA detectable but not quantifiable.

in HBV reactivation, diagnostic escape, and development of HCC. Due to lack of standardized assays, there is limited knowledge on the rate of OHB, especially in HIV patients in Ethiopia. In this study, the prevalence of OHB was found to be 19% (22/115) among anti-HBC positive HIV patients. Our previous study in the same cohort showed a 5.5% of chronic HBV infection among HIV infected patients [15]. Taken together, these data indicate that HIV patients in Ethiopia are at higher risk of HBV-related liver diseases. When compared with other studies, the finding 19% OHB rate was higher than 9.6% and 17.8% prevalence rates in anti-HBc positive HIV patients from India [20, 21] and 10% from Ivory Coast [22]. However, it was lower than 28.1% OHB rate reported from southern Africa [23].

In this study, HBV genotype D was predominant in 18/22 (81%) of OHB/HIV patients. This was consistent with new accumulating studies from the study area [15, 24]. However, previous studies reported HBV genotype A as predominant genotype in Ethiopia [25, 26]. HBV genotype A is prevalent in Uganda, Kenya, and Tanzania [27]. Genotype D is more frequent in

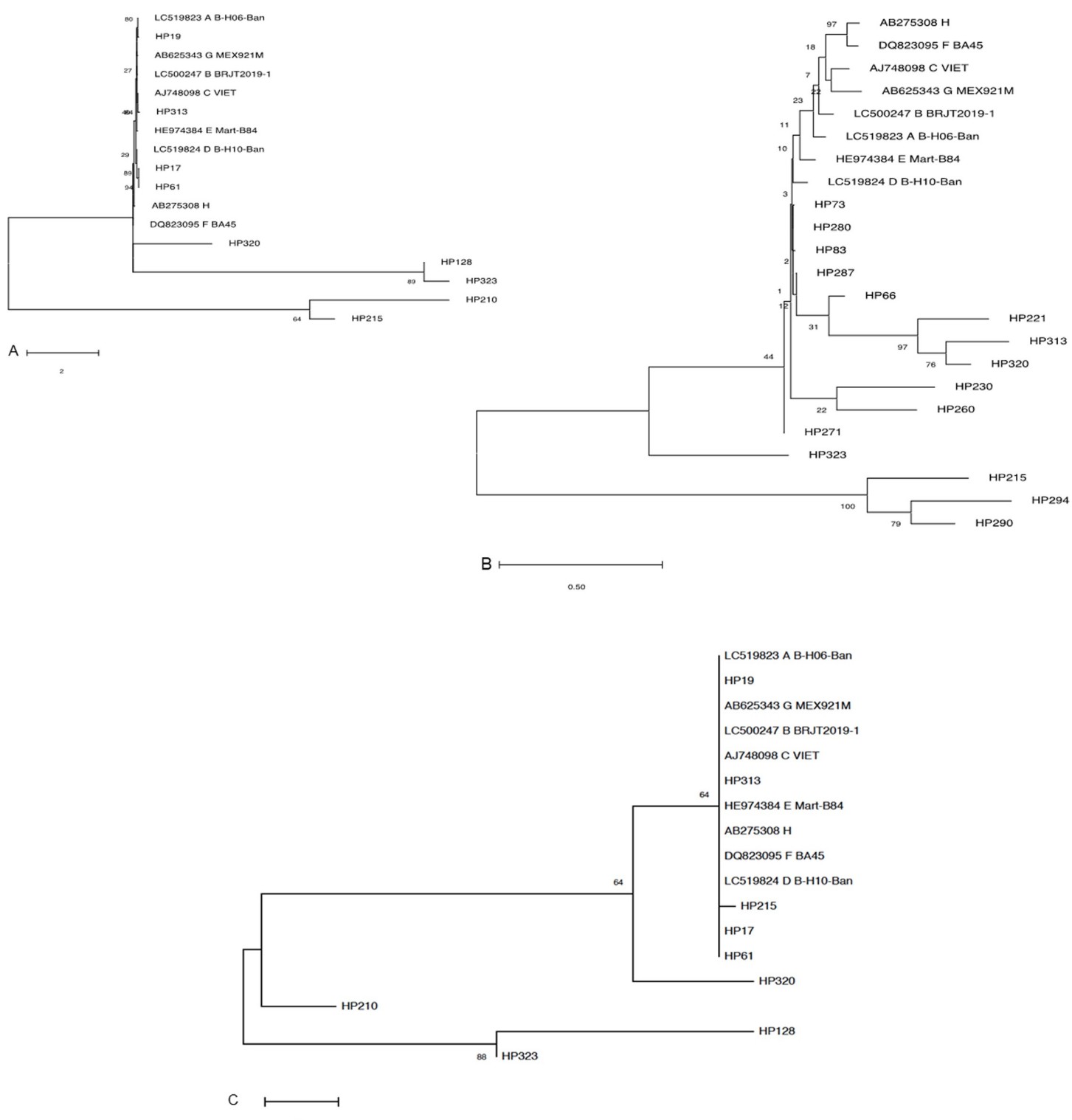

**Fig 2. Phylogenetic analysis of HBV genotypes circulating among occult hepatitis B positive HIV-infected individuals in Northwest Ethiopia.** Maximum-likelihood phylogenetic tree based on A) HBV polymerase, B) pre-core/core, and C) surface genes from OHB/HIV-1 co-infected patients. The bootstrap values based on 1000 replicates are shown next to the branches. The reference genes representing all HBV genotypes (A-H) are available at NCBI GenBank.

Egypt and Sudan [27]. Therefore, geographical proximity of the study area (Gondar) to Sudan and Egypt could explain why genotype D is predominant in the area.

This study also examined the frequency of mutations associated with HCC risk, HBV reactivation, and drug resistance. Entecavir, 3TC and/or TDF associated drug resistance mutations were detected in 23% (5/22) of the patients. This is much higher than the rates reported in other sub-Sahara African countries (<15%) [28, 29]. Interestingly, most of the drug resistance mutations detected in our study were among patients receiving 3TC as the only HBV active agent in the combination antiviral drug. In Ethiopia, HBV is not routinely tested in HIV patients; and HIV patients are treated empirically with antiviral drugs that are also active against HBV (i.e., lamivudine or 3TC), which could select for anti-HBV drug resistance among HBV positive patients [30]. The finding high rate of anti-HBV drug resistance mutation could relate to this practice. Of note, 3TC and entecavir resistance mutations rtM250L and rtM204I were also detected in ART naïve patient, and TDF related mutation (rtA194T) was noted in patients with no history of this drug. These could be due to infection by drug resistant strains circulating in high-risk group, and/or due to *de novo* mutations. Overall, our findings underscore the need to screen all HIV patients for HBV prior to treatment initiation.

Mutations associated with HBV reactivation risk and HCC were detected in 14% and 45% of OHB positive HIV patients respectively. These observations clearly indicate that OHB is a great concern among HIV infected patients as onset of the disease and initial development of liver damage may go undetected for many years and calls for integrating OHB screening (anti-HBc and HBV DNA) for proper management of liver disease in HIV patients.

In conclusion, we found a high rate of occult hepatitis B in anti-HBc positive and HBsAg negative HIV patients. Furthermore, high rates of mutations associated with HBV reactivation, drug resistance, and HCC risk were detected in these patients. To our knowledge, this is the first study to report of these mutations in an OHB/HIV cohort and these would not have been detected through standard HBsAg screening. As such, OHB screening should be performed in HIV positive patients for better management and prevention of HBV-related liver disease.

## Acknowledgments

We would like to thank Ms. Gurmeet Bindra, University of Calgary Liver Unit Biobank Coordinator. We are grateful to the global health office of the University of Calgary for its assistance with coordination of various activities. We would also like to thank the University of Gondar Teaching Hospital physicians, ART staff, and counselling nurses for their kind cooperation through the data collection process.

## Author Contributions

**Conceptualization:** Nishi H. Patel, Vanessa Meier-Stephenson, Meaza Genetu, Debasu Damtie, Ebba Abate, Shitaye Alemu, Yetework Aleka, Guido Van Marle, Kevin Fonseca, Carla S. Coffin, Tekalign Deressa.

**Data curation:** Nishi H. Patel, Vanessa Meier-Stephenson, Meaza Genetu, Debasu Damtie, Ebba Abate, Shitaye Alemu, Yetework Aleka, Guido Van Marle, Kevin Fonseca, Carla S. Coffin, Tekalign Deressa.

**Formal analysis:** Nishi H. Patel, Vanessa Meier-Stephenson, Guido Van Marle, Carla S. Coffin, Tekalign Deressa.

**Funding acquisition:** Shitaye Alemu, Guido Van Marle, Carla S. Coffin, Tekalign Deressa.

**Investigation:** Vanessa Meier-Stephenson, Meaza Genetu, Debasu Damtie, Ebba Abate, Shitaye Alemu, Yetework Aleka, Guido Van Marle, Kevin Fonseca, Carla S. Coffin, Tekalign Deressa.

**Methodology:** Vanessa Meier-Stephenson, Meaza Genetu, Debasu Damtie, Ebba Abate, Shitaye Alemu, Yetework Aleka, Guido Van Marle, Kevin Fonseca, Carla S. Coffin, Tekalign Deressa.

**Project administration:** Carla S. Coffin, Tekalign Deressa.

**Writing – original draft:** Nishi H. Patel.

**Writing – review & editing:** Nishi H. Patel, Vanessa Meier-Stephenson, Meaza Genetu, Debasu Damtie, Ebba Abate, Shitaye Alemu, Yetework Aleka, Guido Van Marle, Kevin Fonseca, Carla S. Coffin, Tekalign Deressa.

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
