## [Decision Letter · Decision Letter 0]

28 Aug 2020

PONE-D-20-19293

Prevalence and genetic variability of occult hepatitis B virus in a human
immunodeficiency virus positive patient cohort in Gondar, Ethiopia

PLOS ONE

Dear Dr. Deressa,

Thank you for submitting your manuscript to PLOS ONE. After careful consideration, we
feel that it has merit but does not fully meet PLOS ONE’s publication criteria as it
currently stands. Therefore, we invite you to submit a revised version of the
manuscript that addresses the  minor but important points raised during the review
process.

If you would like to make changes to your financial disclosure, please include your
updated statement in your cover letter. Guidelines for resubmitting your figure
files are available below the reviewer comments at the end of this letter.

We look forward to receiving your revised manuscript.

Kind regards,

Isabelle Chemin, PhD

Academic Editor

PLOS ONE

Journal Requirements:

2. Please provide additional details regarding participant consent.

In the ethics statement in the Methods and online submission information, please
ensure that you have specified (i) whether consent was informed and (ii) what type
you obtained (for instance, written or verbal, and if verbal, how it was documented
and witnessed).

If the need for consent was waived by the ethics committee, please include this
information.

3. Your ethics statement must appear in the Methods section of your manuscript. If
your ethics statement is written in any section besides the Methods, please move it
to the Methods section and delete it from any other section. Please also ensure that
your ethics statement is included in your manuscript, as the ethics section of your
online submission will not be published alongside your manuscript.

Reviewers' comments:

Reviewer's Responses to Questions

**Comments to the Author**

1. Is the manuscript technically sound, and do the data support the conclusions?

Reviewer #1: Yes

2. Has the statistical analysis been performed
appropriately and rigorously? 

Reviewer #1: Yes

3. Have the authors made all data underlying the
findings in their manuscript fully available?

Reviewer #1: Yes

4. Is the manuscript presented in an intelligible
fashion and written in standard English?

Reviewer #1: Yes

5. Review Comments to the Author

Reviewer #1: The authors described the prevalence of occult hepatitis B infection
(OBI) in HIV infected patients, as well as their genotypes and mutations in S, P and
pre-C-C regions. The study was conducted on 308 HIV + patients’ in Gondar, Ethiopia.
The authors had already assessed the prevalence of HBsAg in this same cohort in
2017. OBI was identified in 291 HBsAg negatives HIV infected patients. Obove them
115 (39.5%) were anti-HBc (+) with an OBI prevalence over 19% (22/115 patients).

Male gender groups seem to be the most affected (prevalence not calculated by the
authors) and are all of D genotypes.

Genotype D has been reported to be predominant. Mutations associated with HBV
reactivation were detected in 3/22 OBI patients. The most common resistant mutation
rtM204V / I (lamivudine), affecting the C domain of the polymerase by transforming
the YMDD motif into YVDD or YIDD was found in one treatment-naive patient. The
authors show the importance of diagnosing OBI in HIV infected patients.

Material and methods

Line 106-107: Patient recruitment period was not specified.

Lines 174-175; 196-197: legends of Figures 1 and 2 are not at their appropriate
place. They should be placed under the corresponding figures.

Primers sequences and amplified fragments size were not specified in the
methodology.

Statistical analyzes:

the p value should be specified.

Results:

Line 146-154 and table 1: We understand that it is the same cohort, but the same
results have already been published by the same team in 2017 as well as all the
results of table 1, which already appeared in table 1 of the ref 15. A summary of
these results might be sufficient.

The authors should focus on new findings specific to the sub-cohort where OBI is
reported.

Image quality of phylogenetic tree could be improved. It will help to differentiate
the patients’ sequences from references. Highlighting the patients’ sequence when
editing the phylogenetic tree would improve clarity.

- It would be interesting to see viral loads results.

- Was co-infection with hepatitis C investigated?

- Did the authors find deletions in S region?

Ligne 223: Cote d’ Ivoire (french) = Ivory coast

References

Reference 2: authors list doesn’t appear on the reference 2.

6. PLOS authors have the option to publish the peer
review history of their article (what does this mean?). If published, this will
include your full peer review and any attached files.

If you choose “no”, your identity will remain anonymous but your review may still be
made public.

**Do you want your identity to be public for this peer review?** For
information about this choice, including consent withdrawal, please see our
Privacy Policy.

Reviewer #1: No

---

## [Author Response · Author response to Decision Letter 0]

27 Oct 2020

I have uploaded a point-by-point response to the reviewers comments together with the
cover letter, please find.

to the reveiwers_Oct, 2020_.docx
---

## [Editor Report · Decision Letter 1]

5 Nov 2020

Prevalence and genetic variability of occult hepatitis B virus in a human
immunodeficiency virus positive patient cohort in Gondar, Ethiopia

PONE-D-20-19293R1

Dear Dr. Deressa,

We’re pleased to inform you that your manuscript has been judged scientifically
suitable for publication and will be formally accepted for publication once it meets
all outstanding technical requirements.

Kind regards,

Isabelle Chemin, PhD

Academic Editor

PLOS ONE
---

## [Editor Report · Acceptance letter]

9 Nov 2020

PONE-D-20-19293R1 

Prevalence and genetic variability of occult hepatitis B virus in a human
immunodeficiency virus positive patient cohort in Gondar, Ethiopia 

Dear Dr. Deressa:

I'm pleased to inform you that your manuscript has been deemed suitable for
publication in PLOS ONE. Congratulations! Your manuscript is now with our production
department. 

Kind regards, 

on behalf of

Mrs Isabelle Chemin 

Academic Editor

PLOS ONE